# Liraglutide Inhibits Hepatitis C Virus Replication Through an AMP Activated Protein Kinase Dependent Mechanism

**DOI:** 10.3390/ijms20184569

**Published:** 2019-09-14

**Authors:** Mei-Yueh Lee, Wei-Chun Chen, Wei-Hao Hsu, Szu-Chia Chen, Jin-Ching Lee

**Affiliations:** 1Division of Endocrinology and Metabolism, Department of Internal Medicine, Kaohsiung Medical University Hospital, Kaohsiung Medical University, Kaohsiung 80708, Taiwan; 2School of Medicine, College of Medicine, Kaohsiung Medical University, Kaohsiung 80708, Taiwan; 3Graduate Institute of Medicine, College of Medicine, Kaohsiung Medical University, Kaohsiung 80708, Taiwan; 4Department of Internal Medicine, Kaohsiung Municipal Siaogang Hospital, Kaohsiung Medical University, Kaohsiung 81267, Taiwan; 5Division of Nephrology, Department of Internal Medicine, Kaohsiung Medical University Hospital, Kaohsiung Medical University, Kaohsiung 80708, Taiwan; 6Department of Biotechnology, College of Life Science, Kaohsiung Medical University, Kaohsiung 80708, Taiwan; 7Graduate Institute of Natural Products, College of Pharmacy, Kaohsiung Medical University, Kaohsiung 80708, Taiwan; 8Drug Development and Value Creation Research Center, Kaohsiung Medical University, Kaohsiung 80708, Taiwan

**Keywords:** hepatitis C virus, liraglutide, GLP-1, diabetes mellitus, AMPK

## Abstract

Insulin resistance and diabetes are both associated with chronic hepatitis C virus (HCV) infection, and the glucagon-like peptide-1(GLP-1) receptor agonist, liraglutide, is a common therapy for diabetes. Our aim was to investigate whether liraglutide treatment can inhibit HCV replication. A cell culture-produced HCV infectious system was generated by transfection of in vitro-transcribed genomic JFH-1 ribonucleic acid (RNA) into Huh-7.5 cells. Total RNA samples were extracted to determine the efficiency of HCV replication. The Ava5 cells were treated with liraglutide and cell viability was calculated. A Western blot analysis of the protein expression was performed. The immunoreactive blot signals were also detected. Liraglutide activated GLP-1 receptors in the HCV infectious system, and inhibited subgenomic HCV RNA replication in the HuH-7.5 cells. The Western blot analysis revealed both HCV protein and replicon RNA were reduced after treatment with liraglutide in a dose-dependent manner. Liraglutide decreased the cell viability of HCV RNA at an optimum concentration of 120 μg/mL, activated the 5′ adenosine monophosphate-activated protein kinase (AMPK) and the phosphorylated- transducer of regulated cyclic adenosine monophosphate (CAMP) response element-binding protein 2 (TORC2), thereby decreasing the cell viability of phosphoenolpyruvate carboxykinase (PEPCK) and G6pase RNA Therefore, we conclude that liraglutide can inhibit HCV replication via an AMPK/TORC2-dependent pathway.

## 1. Introduction

Currently, 415 million adults have diabetes worldwide, a number that is predicted to increase by 55% by 2040 to 642 million [1]. The global prevalence of hepatitis C virus (HCV) is estimated to be 2.5% (177.5 million adults infected with HCV), a global viremic rate of 67% (118.9 million cases positive for HCV RNA) ranging from 64.4% in Asia to 74.8% in Australia, 1.3% in the Americas to 2.9% in Africa [2].

Chronic HCV (CHC) infection can lead to liver cirrhosis, hepatocellular cancer, liver failure, and even death [3], especially in human immunodeficiency virus (HIV) positive patients on active antiretroviral therapy [4]. According to evidence on the natural history of HCV infection, 6% of patients will develop hepatic decompensation because of cirrhosis, 4% will develop hepatocellular carcinoma, and 3% to 4% will die or require a liver transplantation [5]. CHC infection has also been associated with multi-system manifestations beyond the liver which can occur irrespective of the stage of the liver disease including type 2 diabetes mellitus and other systemic diseases [6,7]. Even though the incidence of HCV infection has decreased in the developed world, mortalities due to the co-morbidities with HCV infection like metabolic syndrome associated diabetes mellitus, cardiovascular, and kidney diseases are expected to continue to increase over the next 20 years [8]. Therefore, even though many studies suggest that HCV infection could be eradicated in the next 20 years via therapeutic approaches [9,10], a clear comprehension of HCV infection is needed to allow the evolution of new therapeutic approaches in preventing new infections. 

CHC infection has also been associated with high prevalence rates of insulin resistance (IR) [11,12] and diabetes mellitus [11,13]. Moreover, patients with CHC infection have been reported to have a 2.3-fold higher risk of having type 2 diabetes mellitus [14]. The molecular mechanisms of IR in patients with HCV infection are complex, and involve interactions between viral and host-related factors. The host-related factors involve the interplay between both environmental and genetic factors, which interact with HCV infection to induce IR. In addition, patients with hepatic IR and CHC infection are associated with steatosis and higher expressions of intrahepatic inflammatory cytokines and reactive oxygen species, which can potentially interfere with insulin signaling [15]. 

The primary goal of CHC infection treatment is the permanent elimination of the virus. Even if eradication of the virus is unsuccessful, anti-viral treatment can slow the progression and complications of the disease [16,17]. The gold-standard treatment for CHC infection is currently a combination of ribavirin (RBV) and pegylated interferon alpha (PEG-interferon (IFN) α). In addition, the use of antiviral therapy based on a combination of PEG-IFN α-2a or 2b and RBV has been reported to achieve virus eradication in 40–50% of cases [18]. However, side effects have been reported in nearly 80% of patients receiving combination therapy of PEG-IFN α and RBV for CHC infection [19]. Obstacles remain with regards to the implementation of new pan-genotypic, RBV-free direct-acting antiviral (DAA) regimens [20]. DAAs are a powerful tool in the fight against CHC infection, and their efficacy has led to many associations recommending them as a first-line treatment [21]. There are four classes of hepatitis C treatment combine in different ways to make up DAAs: (1) NS3/4A protease inhibitors (PI), (2) nucleoside and nucleotide NS5B polymerase inhibitors, (3) NS5A inhibitors, and (4) non-nucleoside NS5B polymerase inhibitors, which directly target the HCV in different mechanisms to stop it from replicating itself and has done so remarkably well that they hold the promise for a brighter future with hepatitis C infected people. DAAs promise therapy with shorter treatment duration, greater cure rates, and fewer side effects [21]. However, despite their promising results, their cost limits their widespread use. Taken together, these findings imply that effective HCV eradication will require significant investment for screening, prevention, and treatment programs to identify patients with CHC infection in the general population and in specific high-risk groups such as those with diabetes mellitus. 

HCV-infected Huh 7.5 cells have been reported to show high levels of peroxisome proliferator-activated receptor-gamma coactivator alpha (PGC1-α) and elevated glucose production [22]. Pavone et al. have found that with diabetic patients infected with CHC who were treated with interferon-free DAA-based therapy, the glucose and hemoglobin A1C (HbA1c) levels were significantly lowered during the course of treatment [23]. However, more evidence is needed to confirm the relationship between type 2 diabetes mellitus and CHC infection.

Liraglutide is a long-acting glucagon-like peptide-1 (GLP-1) receptor agonist, which binds to the same receptors as the endogenous metabolic hormone GLP-1. It has been shown to stimulate insulin secretion, and it was developed for the treatment of type 2 diabetes. In a previous animal study, liraglutide was shown to enhance hepatic insulin sensitivity but not in skeletal muscles under normal glucose tolerance conditions. However, in an insulin-resistant state, liraglutide was shown to improve both hepatic and muscle insulin resistance, then improve fatty liver [24]. In this study, we aimed to investigate whether liraglutide can inhibit HCV replication through the same mechanism.

## 2. Results

### 2.1. Liraglutide Inhibited HCV RNA Replication in a Dose-Dependent Manner

The expression of nonstructural 5B (NS5B) viral protein decreased in a dose-dependent manner with liraglutide treatment, most prominently at 60 and 120 μg/mL, and this inhibitory effect on HCV RNA replication was nearly the same as with the comparator IFN without interfering with the relative cell viability (Figure 1).

### 2.2. Liraglutide Activated GLP-1 Receptors in the HCV Replicon System 

Liraglutide activated GLP-1 receptors in the HCV replicon system, and increased the expression of phosphorylated 5′ AMP-activated protein kinase (p-AMPK) protein in a dose- and time-dependent manner at 120 μg/mL (Figure 2).

### 2.3. Effect of AMPK Inhibition on NS5B with Liraglutide Pretreatment 

AMPK inhibition reversed the inhibitory effect on HCV RNA replication with liraglutide 120 μg/mL treatment in a dose-dependent manner. The expression of NS5B viral protein increased significantly at a dose of 20 μM of the AMPK inhibitor after pretreatment with liraglutide 120 μg/mL, and decreased the protein expression of p-AMPK at the same dose (Figure 3A,B). Similar results were observed with AMPK short hairpin RNA (shRNA) transfection. AMPK shRNA significantly reduced AMPK expression and liraglutide-induced p-AMPK expression, which was comparable with the restoration of HCV protein synthesis (Figure 3C). Similarly, the recovery of HCV RNA levels coincided with decreasing AMPK protein levels by AMPK shRNA in liraglutide-treated cells (Figure 3D).

### 2.4. Effects of Liraglutide on p-TORC2, Glucose 6 Phosphatase (G6Pase) RNA, and Phosphoenolpyruvate Carboxykinase (PEPCK) RNA

Liraglutide treatment significantly increased the protein expression of p-TORC-2 at a dose of 120 μg/mL and dose-dependently reduced the RNA level of G6Pase and PEPCK which is necessary for HCV replication (Figure 4A–C). To determine the effect of liraglutide-induced phosphorylation of TORC2 against HCV, the Ava5 cells were treated with liraglutide and combined with AMPK inhibitor for 3 days. The AMPK inhibitor significantly reduced liraglutide-induced phosphorylation level of TORC2 (Figure 4D), similarly in CRE reporter-based assay and the expression of G6Pase and PEPCK (Figure 4E,F), which is comparable with the restoration of HCV protein synthesis. 

## 3. Discussion

In this study, liraglutide activated GLP-1 receptors in the HCV replicon system through the AMPK pathway by activating TORC2 and inhibiting the transcription of target genes such as PEPCK and G6Pase (Figure 5). Several intracellular pathogens have been shown to manipulate the AMPK/insulin-mammalian target of rapamycin (mTOR) pathway during the state of infection via either directly targeting AMPK or mTOR or by targeting the upstream or downstream pathways, including HCV, human cytomegalovirus (HCMV), Leishmania, Francisella, Rift Valley fever virus (RVFV), and simian virus 40 (SV40). These effects resulted an increase in the activity of mTOR but decrease in AMPK via HCV infection [25].

The genomic size of HCV is approximately 9.6 kb, and includes 10 different structural proteins (core, E1, E2, p7) and non-structural proteins (NS2–NS5), which help the virus to replicate and damage the host machinery. Various in vitro studies have demonstrated chronic background inflammation and an increase in mitochondrial reactive oxygen species (ROS) in HCV infection. Of these proteins, Core, NS-3, and NS-5 were shown to trigger oxidative stress responses and were mainly implicated in IR, with the core protein being involved in capsid formation, NS-3 inducing helicase and proteolytic activity, NS-5A downregulating IFN-stimulated genes, and NS-5B being an RNA polymerase [26,27]. NS-5B participates in the replication of viral HCV RNA by using the RNA strand positive for the virus as a template to catalyze the polymerization of ribonucleoside triphosphates during RNA replication [28]. Human monocytes incubated with varying HCV proteins have shown that NS3 selectively generated ROS by activation of NADPH oxidase and Nox2 [29]. Human hepatoma Huh-7 cells transfected with an NS5A vector have demonstrated an elevation of ROS with resulting activation of NF-KB and STAT-3 pathways [30] that then led to the release of various array of cytokines, including IL-6, IL-8, TNFα, and TGFβ. The structural core protein has also been shown to cause an increase in ROS, mitochondrial dysfunction, and endoplasmic reticulum (ER) stress by feasibly profuse glutathione stores and ER chaperones during viral replication [31,32].

The vigorous inflammatory response to HCV is suspected to be central to the development of peripheral and hepatic IR in chronic HCV infection, primarily due to via interference in the insulin signaling pathway. Several studies have confirmed that TNFα can directly intervene with insulin signaling in HCV patients [33,34].

There are studies shown that HCV has direct effects on insulin signaling. HCV infection of liver cells can lead to (1) decreased IR auto phosphorylation; (2) decreased IR substrate-1(IRS-1) activation due to elevated serine-phosphorylation of IRS-1; (3) diminished IRS-1 levels due to expanded ubiquitin-mediated proteasomal degradation caused by mTOR upregulation and suppressor of cytokine signaling (SOCS 3/7); (4) decreased protein kinase B (PKB) activity due to increased threonine-phosphorylation of PKB; (5) reduced glucose transporter type 4 (GLUT4) expression; and (6) increased gluconeogenic enzymes glucose-6-phosphatase and phosphoenolpyruvate carboxykinase 2 (GC6P and PCK2) [35,36,37,38].

AMPK plays a key molecular role in energy homeostasis at both a cellular and whole-body level [39]. Activated AMPK has been shown to stimulate catabolic pathways (glycolysis, fatty acid oxidation, and mitochondrial biogenesis) and inhibit anabolic pathways (gluconeogenesis, glycogen, fatty acid and protein synthesis). Liver kinase B1 (LKB1) and calcium/calmodulin-dependent protein kinase kinase β (CaMKKβ, also known as CaMKK2) have been identified as the two major upstream kinases capable of phosphorylating Thr^172^ in mammalian cells [40,41]. AMPK phosphorylated on Thr^172^ can be dephosphorylated by protein phosphatases. In mammalian cells, AMPK is activated by an increase in the AMP:ATP and ADP:ATP ratio, that occurs in response to a fall in ATP levels [42]. AMP and ADP binding to the γ subunit of AMPK promote phosphorylation of Thr^172^, and AMP and ADP protect pThr^172^ from dephosphorylation. As well as regulating the phosphorylation status of Thr^172^, AMP, but not ADP, allosterically activates AMPK. The metabolic effects of GLP-1 are well understood. GLP-1 rapidly increases cyclic AMP and intracellular calcium in pancreatic β-cells, which in turn stimulates insulin exocytosis in a glucose-dependent manner, hence promotes glucose homeostasis [43].

AMPK has also been reported to intervene in the metabolic effects of cannabinoids and hormones including insulin, glucocorticoids, adiponectin, leptin, and ghrelin [44]. Hyperglycemia results if IR occurs when insulin cannot increase glucose uptake or suppress glucose production in the liver. The role of AMPK should be considered in this condition. Activated AMPK promotes catabolic pathways and suppress energy-consuming anabolic processes, whereas insulin stimulates the synthesis of glycogen, proteins, and lipids. Activated AMPK has been shown to upregulate insulin receptor substrate-1 [45] through inhibiting the mTOR pathway [46], so that improving the profile of insulin sensitivity. Both insulin and activated AMPK have been reported to suppress the expressions of the gluconeogenic enzymes PEPCK and G6Pase [47], and phosphorylation of AMPK results in the translocation of transcriptional coactivator TORC2 to the cytoplasm, which shows the interaction with cAMP response element-binding protein (CREB) to active its dependent gene expression and has been identified as an important regulator of gluconeogenesis in the livers of mammals [48], thus repressing the expression of TORC2-targetted enzymes.

HCV infection has been shown to promote the expression of gluconeogenic genes thereby enhancing IR. It has also been shown that HCV can promote fatty acid synthesis through the upregulation of lipogenic gene sterol regulatory element binding protein 1c, which then promotes the transcriptional activation of other lipogenic genes such as acetyl CoA carboxylase, ATP citrate lyase, and hydroxymethylglutaryl CoA reductase [49]. The sterol regulatory element binding proteins (SREBPs), transcription factors activate genes encoding enzymes of cholesterol and fatty acid biosynthesis, were regulated with AMPK and induced upon HCV infection [50,51]. Besides, the viral proteins activated lipid alteration including lipid droplet formation and autophagosome synthesis promote the viral replication and are associated with the production of efficient infectious virus particles [52]. The PGC1-α activation by HCV infection leading to gluconeogenic genes expression, which is also related to viral replication and assembly [53]. The interaction between CREB and TORC2 is responsible for PGC1-α activation [54], thereby we suggest that the reduction expression of G6Pase and PEPCK expression by liraglutide may be due to the disruption of TORC2 translocation leading PGC1-α inhibition**.** A recent study also reported that in an HCV-infected cell line Huh.8, PEPCK which is a key regulator of gluconeogenesis and cellular lipids, was strongly upregulated in association with HCV NS-5A expression. This suggests that NS-5A may be involved in gluconeogenesis and imbalances in glucose metabolism [55].

Two gluconeogenic enzymes which could be targets for diabetic treatment are PEPCK, which converts oxaloacetate to PEP, and G6Pase, which catalyzes the conversion of glucose-6-phosphate to glucose. The expressions of these enzymes have been shown to be highly regulated by glucagon, insulin, and gluconeogenic flux [56]. In addition, the expression of PEPCK has been shown to be dysregulated in diabetes, and a seven-fold increase in the expression of PEPCK has been shown to result in hyperglycemia in mice [57]. PEPCK is consequently thought to be a rate-limiting enzyme for gluconeogenesis, and is implicated as a potential target to reduce glucose production in the liver and blood glucose levels. In the current study, liraglutide inhibited PEPCK and G6Pase through the activation of TORC2 via the AMPK pathway.

HCV patients with diabetes should be treated according to the American Diabetes Association (ADA) guidelines [58]. Management of diabetic patients with hepatic problems, however, is complicated by the hepatic effects of drug metabolism, including drug–drug interactions, and the potential hepatotoxicity of oral antidiabetic agents [59]. GLP-1 is a gut-derived incretin hormone that stimulates insulin secretion. GLP-1 is stimulated postprandially and is mainly secreted from intestinal L-cells and neurons in the caudal region of the nucleus of the solitary tract, which then releases GLP-1 into the hypothalamus to control appetite. Activation of GLP-1R by intestinal GLP-1 in the pancreas can stimulate the secretion of insulin, increase insulin sensitivity, and inhibit the release of glucagon. An increase in the expression of insulin along with ameliorated insulin signaling in glucoregulatory organs such as the liver, muscle and adipose tissue can intensify glucose uptake and metabolism, and consequently reduce hepatic gluconeogenesis and improve steatosis. GLP-1 has also been reported to improve inflammatory immune responses via directly addressing immune cells and consequently improving systemic or metabolic inflammation and steatohepatitis. Dipeptidyl peptidase-4 (DPP-4) inhibitors enhanced GLP-1 levels. Itou et al. has shown that serum GLP-1 levels were significantly reduced in HCV patients compared to control and hepatitis B virus infected groups; and that DPP-4 expression was significantly elevated in the ileum, liver, and serum in the HCV group. According to the authors, HCV related glucose intolerance maybe due to the involvement of alteration in the expression of GLP-1 [60]. Recent studies on GLP-1 have also reported a deceleration in the progression of non-alcoholic fatty liver disease by direct effects on lipid metabolism in hepatocytes, and on liver inflammation. DPP-4 inhibitors may also affect hepatic pathways of fat removal [61,62]. A case-control study assessed the efficacy and safety of DPP-4 inhibitors in type 2 diabetic patients with HCV and showed HbA1C reduction without side effects [63]. However, dipeptidyl peptidase-4 (DPP-4), which is ubiquitously expressed on most blood and tissue cells, has been shown to rapidly inactivate the GLP-1 secreted by L-cells, highlighting the potential application of proteolytically stable GLP-1 mimetics or DPP-4 inhibitors to treat type 2 diabetes, metabolic syndrome, and even nonalcoholic steatohepatitis [64].

In conclusion, HCV is a multifaceted disorder involving various cellular and viral factors for disease progression. CHC infection is likely to induce IR through glucose imbalance and hepatic gluconeogenesis. Liraglutide, a GLP-1 receptor agonist, can inhibit the replication of HCV via an AMPK/TORC2-dependent pathway, which is an important pathway in gluconeogenesis.

## 4. Materials and Methods

### 4.1. Cell Cultures and Reagents

Naïve Huh7 and Ava5 cells, an Huh7 cell line harboring HCV replicon cells [65] were maintained in complete Dulbecco’s modified Eagle’s medium (DMEM) supplemented with 10% fetal bovine serum (FBS), 1% non-essential amino acids, and 1% Antibiotic-Antimycotic Solution (Life Technologies Co., Grand Island, NY, USA). Liraglutide was obtained from Novo Nordisk Pharmaceutical Company, Denmark. The AMPK inhibitor, compound C, was obtained from Sigma Aldrich Co. (Sigma, St. Louis, MO, USA). All stock compounds were stored at 4 °C until use, and all of the reactions were diluted with fresh medium.

### 4.2. Virus Infection

A cell culture-produced HCV (HCVcc) infectious system [66] was generated according to the method described by Wakita et al. [67,68] by transfection of in vitro-transcribed genomic JFH-1 RNA into Huh-7.5 cells. The Huh-7 cells were infected with HCV for 8 h at a multiplicity of infection (MOI) of 0.1. Total RNA samples were extracted and subjected to quantitative real-time polymerase chain reaction (qRT-PCR) analysis to determine the efficiency of HCV replication.

### 4.3. Quantitative Real-Time PCR

Total cellular RNA was extracted and purified using a Total RNA Miniprep Purification Kit (GMbiolab Co., Ltd., Taichung, Taiwan). Complementary DNA synthesis was performed using an M-MLV Reverse Transcription System (Promega Biosciences, San Luis Obispo, CA, USA). The relative levels of genes expressions were quantified by qRT-PCR as previous described [69]. Relative expressions were calculated using the comparative Ct method and normalized to the expression of the glyceraldehydes-3-phosphate dehydrogenase (*GADPH)* gene. The sequencing primers for qRT-PCR were used.

### 4.4. Cell Viability

To detect the cytotoxic effect of the liraglutide, a CellTiter 96 Aqueous One Solution Cell Proliferation assay system (Promega, Madison, WI, USA) was used to measure cell viability according to the manufacturer’s protocol. Ava5 cells were seeded in 96-well plates and treated with liraglutide at the indicated concentrations. After 3 days, cell viability was calculated according to absorbance values which were detected at 490 nm.

### 4.5. Western Blot Assay

The total lysate was extracted and protein expression was analyzed using Western blot analysis as previously described [70]. The protein expression was detected with primary antibodies against glyceraldehyde 3-phosphate dehydrogenase (GAPDH) (catalogue number GTX124503, 1:10,000; GeneTex, Irvine, CA, USA), NS5B (catalogue number ab65410, 1:5000; Abcam, Cambridge, MA, USA), GLP-1R (catalogue number orb238545, 1:3000; Biorbyt LLC, San Francisco California, CA, USA), AMPK (catalogue number 2532, 1:3000; Cell Signaling Technology, Inc., Beverly, MA, USA), Phospho-Thr712-AMPK (catalogue number 2535, 1:1000; Cell Signaling), TORC2 (catalogue number GTX31879, 1:1000; GeneTex), or phosphor-Ser171-TORC2 (catalogue number GTX51565, 1:1000; GeneTex). The immunoreactive blot signals were detected using an enhanced chemiluminescence detection kit (Perkin-Elmer, Norwalk, CT, USA). 

### 4.6. Transfection and Reporter Activity Assay

The CRE transactivity luciferase reporter plasmids (pCRE-FLuc) contains CREB binding domains driving firefly luciferase expression, which was used to detect the translocation and transcription activity of CREB. The T-Pro™ reagent was used for transfection following the manufacturer’s instructions (Ji-Feng Biotechnology Co., Ltd., New Taipei, Taiwan). The luciferase activities were analyzed using the Bright-Glo Luciferase assay system (Promega). The cells were co-transfected with 0.1 μg of secreted alkaline phosphatase (SEAP) expression vector (pSEAP) and the transfection efficiency were normalized by SEAP activity. The specific shRNAs targeting AMPK (NM_006252), G6Pase (NM_138387), and PEPCK (NM_002591) were purchased from the National RNAi Core Facility, Institute of Molecular Biology/Genomic Research Center, Academia Sinica, Taiwan. The DNA fragments were confirmed by DNA sequencing.

### 4.7. Statistical Analysis

The results of at least three independent experiments were analyzed and presented as means ± SD. Statistical significance was set as a *p* value < 0.05 or < 0.01 as assessed using the Student’s *t*-test and ANOVA with GraphPad inStat software incorporation.

## Figures and Tables

**Figure 1 ijms-20-04569-f001:**
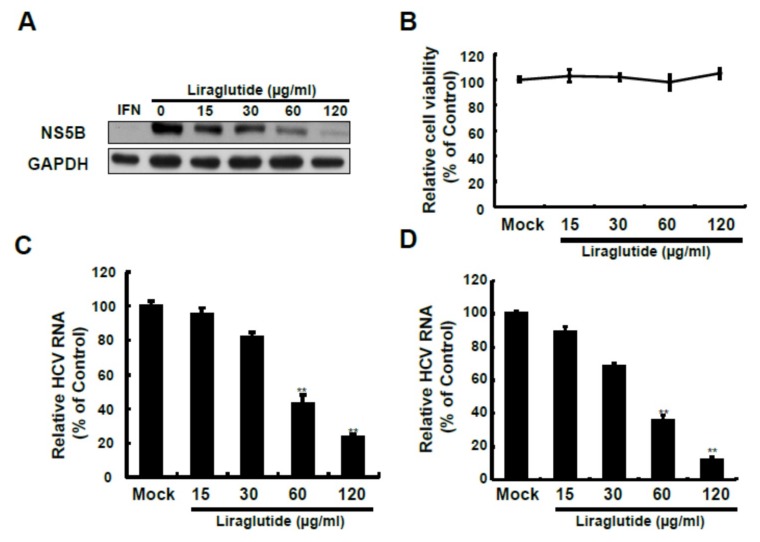
Effects of liraglutide on HCV protein synthesis and RNA replication. (**A**,**B**) Liraglutide inhibits HCV protein synthesis without cytotoxicity. Ava5 cells were treated with liraglutide with indicated concentrations. The cell lysate was extracted and the protein synthesis was analyzed by Western blot using anti-HCV NS5B and anti-GAPDH (loading control) antibodies. Cellular toxicity was determined by the MTS assay. Liraglutide inhibits HCV RNA replication in replicon (**C**) and infectious (**D**) systems. Ava5 and JFH-1-infected Huh-7 cells were treated with liraglutide with indicated concentrations for 3 days. The total RNA was extracted and intracellular HCV RNA levels were normalized by cellular GADPH mRNA expression which were quantified by qRT-PCR. The results shown are presented as the percentage ± standard deviations (SD) from 3 independent experiments. ** *p* < 0.01, as assessed using the Student’s *t*-test for quantification of real-time PCR threshold values from each individual experiment and ANOVA.

**Figure 2 ijms-20-04569-f002:**
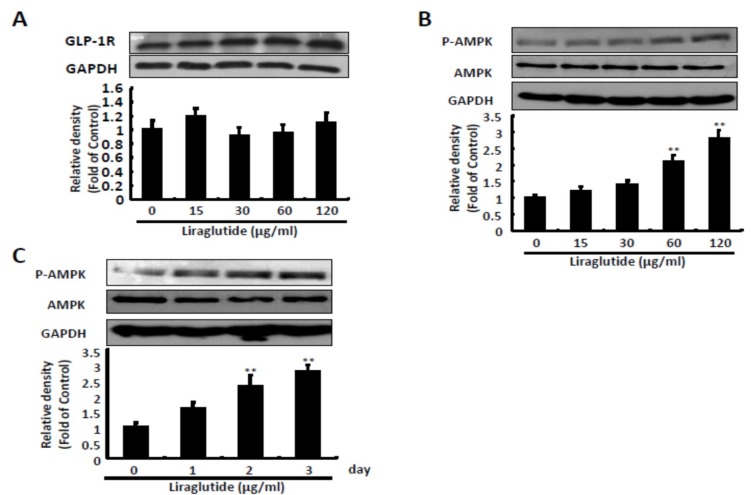
Effects of liraglutide on GLP-1 receptor expression and AMPK phosphorylation. (**A**) Liraglutide makes no significant differences in GLP-1 receptor. Ava5 cells were treated with liraglutide with indicated concentrations for three days. The cell lysate was extracted and the protein synthesis was analyzed by Western blot by using anti-GLP-1R and anti-GAPDH (loading control) antibodies. (**B**,**C**) Liraglutide activated AMPK phosphorylation in a concentration- and time-dependent manner. Ava5 cells were treated with liraglutide with effective antiviral concentrations for 3 days or 120 µg/mL of liraglutide in different lengths of time (1, 2, and 3 days). The cell lysate was extracted and the protein synthesis was analyzed by Western blot using anti-AMPK, anti-phospho-AMPK, and anti-GAPDH (loading control) antibodies. The relative densities of bands are quantified and showed below the Western images. The results shown are presented as the percentage ± standard deviations (SD) from three independent experiments. ** *p* < 0.01, as assessed using the Student’s *t*-test for quantification of relative band intensity from each individual experiment and ANOVA.

**Figure 3 ijms-20-04569-f003:**
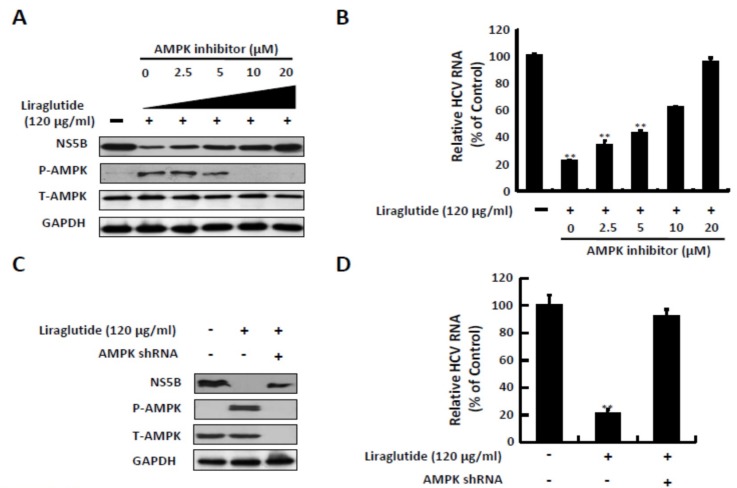
Effects of AMPK inhibitor treatment and shRNA knockdown on the anti-HCV activity of liraglutide. (**A**,**B**) Inhibition AMPK phosphorylation attenuated the anti-HCV activity of liraglutide. Ava5 cells were treated with liraglutide (120 µg/mL) combined with indicated concentrations of AMPK inhibitor (Compound C) for 3 days. (**C**,**D**) Knockdown of AMPK expression attenuated the anti-HCV activity of liraglutide. Ava5 cells were transfected with AMPK shRNA. Subsequently, the transfected cells were treated with liraglutide (120 µg/mL) for 3 days. The cell lysate was extracted and the protein synthesis was analyzed by Western blot using anti-HCV NS5B, anti-AMPK, anti-phospho-AMPK, and anti-GAPDH (loading control) antibodies. The total RNA was extracted and intracellular HCV RNA levels were normalized by cellular GADPH mRNA expression which were quantified by qRT-PCR. The results shown are presented as the percentage ± standard deviations (SD) from three independent experiments. * *p* < 0.05; ** *p* < 0.01, as assessed using the Student’s *t*-test for quantification of real-time PCR threshold values from each individual experiment and ANOVA. + with liraglutide or AMPKshRNAr added; - without liraglutide or AMPK shRNA added.

**Figure 4 ijms-20-04569-f004:**
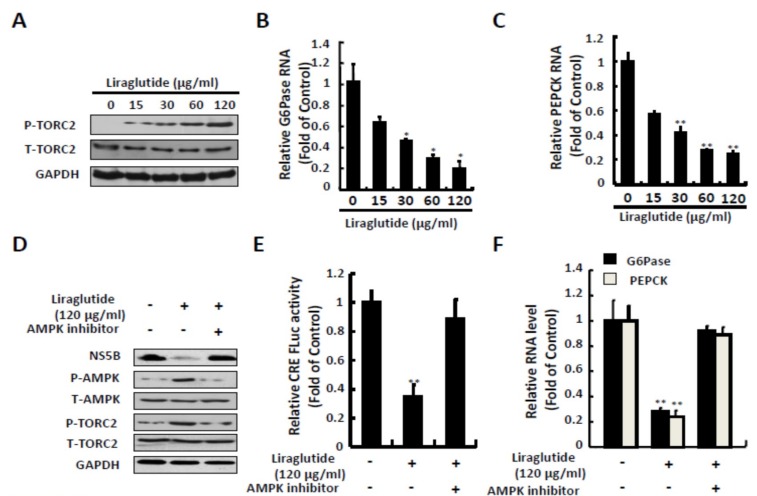
Effects of liraglutide on TORC2 phosphorylation and gluconeogenesis genes expression. (**A**,**B**,**C**) Liraglutide activated TORC2 phosphorylation and suppressed G6Pase and PEPCK expressions in Ava5 cells. Ava5 cells were treated with liraglutide with indicated concentrations for 3 days. (**D**) AMPK inhibitor attenuated liraglutide-induced TORC2 phosphorylation. Ava5 cells were treated with liraglutide (120 µg/mL) combined with AMPK inhibitor (Compound C) for 3 days. (**E**,**F**) AMPK inhibitor attenuated liraglutide-reduced CRE transcription activity and G6Pase and PEPCK expressions. Ava5 cells were transfected with pCRE-FLuc reporter plasmid (1µg) and treated with liraglutide (120 µg/mL) combined with AMPK inhibitor (Compound C) for 3 days. The cell lysate was extracted and the protein synthesis was analyzed by Western blot using anti-Phospho-TORC2, anti-TORC2, anti-HCV NS5B, anti-AMPK, anti-phospho-AMPK, and anti-GAPDH (loading control) antibodies, respectively. The total RNA was extracted and intracellular G6Pase and PEPCK RNA levels were normalized by cellular GADPH mRNA expression which were quantified by qRT-PCR. The relative RNA levels and CRE transactivity were presented as fold changes compared to Ava5 cells in which were presented as 1. The results shown are presented as the fold ± standard deviations (SD) from three independent experiments. * *p* < 0.05; ** *p* < 0.01, as assessed using the Student’s *t*-test for quantification of real-time PCR threshold values and relative luciferase activity from each individual experiment and ANOVA. + with liraglutide or AMPK inhibitor added; - without liraglutide or AMPK inhibitor added.

**Figure 5 ijms-20-04569-f005:**
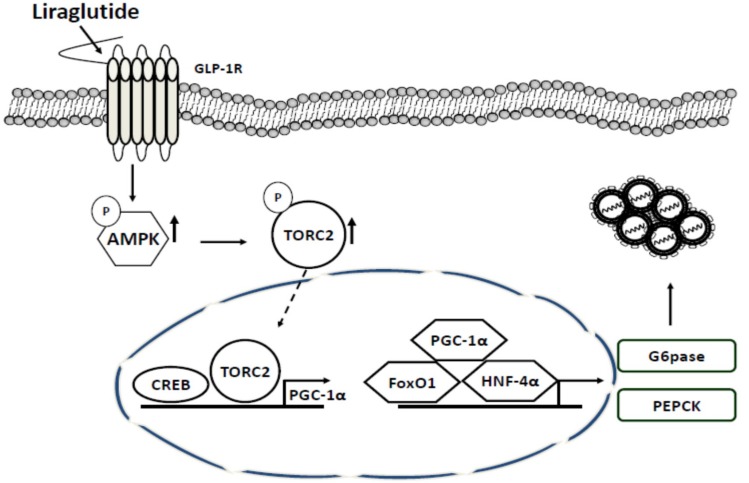
Antiviral action of liraglutide in HCV replication. Liraglutide actuated GLP-1 response to enhance AMPK phosphorylation which phosphorylated TORC2 to depletion nuclear TORC2 to inhibit the transactivation of gluconeogenic genes leading to the suppression of HCV replication.

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
