# Peer review of "Liraglutide Inhibits Hepatitis C Virus Replication Through an AMP Activated Protein Kinase Dependent Mechanism"

_ijms, 2019, doi:10.3390/ijms20184569_

Round 1

Reviewer 1 Report

The manuscript from Lee et al. shows evidence that activation of the glucagon like peptide 1 receptors activates AMPK/TORC2 signalling and suppresses HCV replication. The data are brief and generally correlative, but seem reasonably clear. The impact is limited by the lack of mechanistic detail, but the general conclusion that suppressing synthetic pathways in the cell adversely affects HCV replication seems plausible and consistent with prior research. The discussion could use some significant work to condense it to the necessary details and to better integrate the presented data with past research. As it stands a lot of detail is given about the role of these pathways in diabetes, but not very much is said about the relevant prior HCV research in this area.

One glaring error that is presumably embarrassing to the authors and must be corrected - Fig 1 is missing.

Major points:

Figure 1 is missing, while Figure 2 is shown twice. Reference 55 - McRae et al, has been retracted, perhaps should not be used. Other papers authored by Gulam Waris should also be checked against the published retraction notices, as not all papers were initially retracted (http://www.jbc.org/content/293/52/20011.long; http://www.jbc.org/content/293/52/20010.long; https://journals.plos.org/plosone/article?id=10.1371/journal.pone.0216026)  For the length of the discussion, there are several passages that seem to be lacking in references. e.g. Line 276 -286. It may be that the relevant paper is cited, but it would aid the reader to be reminded which paper(s) is being discussed here. Additioanlly, there is prior research into AMPK and mTOR in relation to HCV that seems to be absent here, despite the fact that it would seem to support the present findings (e.g Mankouri et al.). It is clearly important to understand how the presented data relate to the prior research.

Minor points:

The discussion is long, probably unnecessarily so. A lot of general background is given, but not much is explained in terms of the relationship to HCV replication/pathogenesis. I think it could be streamlined. Methods should include Antibody catalogue numbers. The statistical method used should be reported in the figure legends - the authors report that they used t test and ANOVA, but all figures appear to require ANOVA, so it is unclear where the t tests were applied. Fig. 2C - the GAPDH blot has a clear second band in the 3rd lane. Is this significant? Consistent across all experiments?
Line 57 - I think the authors are arguing that HCV is a significant co-morbidity with diabetes, but the phrasing is not entirely clear.

Author Response

Reviewer 1

The manuscript from Lee et al. shows evidence that activation of the glucagon like peptide 1 receptors activates AMPK/TORC2 signalling and suppresses HCV replication. The data are brief and generally correlative, but seem reasonably clear. The impact is limited by the lack of mechanistic detail, but the general conclusion that suppressing synthetic pathways in the cell adversely affects HCV replication seems plausible and consistent with prior research. The discussion could use some significant work to condense it to the necessary details and to better integrate the presented data with past research. As it stands a lot of detail is given about the role of these pathways in diabetes, but not very much is said about the relevant prior HCV research in this area.

One glaring error that is presumably embarrassing to the authors and must be corrected - Fig 1 is missing.

Major points:

Figure 1 is missing, while Figure 2 is shown twice. 

Response: Please forgive us for this honest mistake. The figure 1 is now replaced with the correct one after careful edit.

Reference 55 - McRae et al, has been retracted, perhaps should not be used. Other papers authored by Gulam Waris should also be checked against the published retraction notices, as not all papers were initially retracted (http://www.jbc.org/content/293/52/20011.long; http://www.jbc.org/content/293/52/20010.long; https://journals.plos.org/plosone/article?id=10.1371/journal.pone.0216026) 

Response: Thank you for your kind reminder. After the search in pubmed, Ref. 55 was withdrawn on Dec. 2018 after 2 years of publication in some reason. It was deleted from our reference now.

 For the length of the discussion, there are several passages that seem to be lacking in references. e.g. Line 276 -286. It may be that the relevant paper is cited, but it would aid the reader to be reminded which paper(s) is being discussed here.

Response: A ref. 44 is added among the lines 272-273 (from original line 276-286)

Additionally, there is prior research into AMPK and mTOR in relation to HCV that seems to be absent here, despite the fact that it would seem to support the present findings (e.g Mankouri et al.). It is clearly important to understand how the presented data relate to the prior research.

Response: Yes, Mankouri et al research can fully support our present finding. We added this article as our reference 50. Thank you for your good recommendation.

Minor points:

The discussion is long, probably unnecessarily so. A lot of general background is given, but not much is explained in terms of the relationship to HCV replication/pathogenesis. I think it could be streamlined. 

Response: Thank you for your comments. We already deleted line 258-265 , 280-290 and 321-325 to make our discussion more condense.

Methods should include Antibody catalogue numbers.

Response: As suggested by reviewer, we added the Antibody catalogue numbers in the section of Materials and methods.

The statistical method used should be reported in the figure legends - the authors report that they used t test and ANOVA, but all figures appear to require ANOVA, so it is unclear where the t tests were applied.

Response: As suggested by reviewer, we added the statistical method in figure legends. The t test was used for quantification of real-time PCR cycle threshold values from each individual experiment.

Fig. 2C - the GAPDH blot has a clear second band in the 3rd lane. Is this significant? Consistent across all experiments?

Response: The clear second band in the 3rd lane of GAPDH blot is the non-specific reaction. We provided the original image data and suggested that minor bands have no significant effects on our results.

Line 57 - I think the authors are arguing that HCV is a significant co-morbidity with diabetes, but the phrasing is not entirely clear. 

Response: Line 57 is rephrase now with comorbidities of HCV were added.

Reviewer 2 Report

This is an interesting study demonstrating that activation of the Glp1 receptor is linked to the activation of signal transduction through mTor resulting in a reduction of HCV replication. The paper will require a major edit since the figure legends do not correspond to the data and therefore I had a difficult time in interpreting the data. For example, the legend for Fig 1 states that there should be a panel D but there is no corresponding data. After a careful edit I can re-review the paper.

Author Response

This is an interesting study demonstrating that activation of the Glp1 receptor is linked to the activation of signal transduction through mTor resulting in a reduction of HCV replication. The paper will require a major edit since the figure legends do not correspond to the data and therefore I had a difficult time in interpreting the data. For example, the legend for Fig 1 states that there should be a panel D but there is no corresponding data. After a careful edit I can re-review the paper.

Response to Reviewer:

Please forgive us for this honest mistake. The figure 1 is now replaced with the correct one after careful edit. Please reconsider to review our manuscript and continue to give us our prestigious comments.

Round 2

Reviewer 1 Report

The authors have addressed my concerns. I have noted a couple of minor points below.

Minor points:

In Fig 1, I think C refers to the replicon system and D to the infection, but this should be made clear in the legend.

The description of the statistical tests is still a little confusing to me. The phrase “as assessed using the Student’s t-test for quantification of real-time PCR threshold values from each individual experiment and ANOVA.” has been added to all legends, including those for figures where no qPCR is shown. Is it simply that all qPCR analyses used T test, while other assays, e.g. FLuc and western blot densitometry, used ANOVA?

Author Response

In Fig 1, I think C refers to the replicon system and D to the infection, but this should be made clear in the legend.

The description of the statistical tests is still a little confusing to me. The phrase “as assessed using the Student’s t-test for quantification of real-time PCR threshold values from each individual experiment and ANOVA.” has been added to all legends, including those for figures where no qPCR is shown. Is it simply that all qPCR analyses used T test, while other assays, e.g. FLuc and western blot densitometry, used ANOVA?

Our response: As suggested by reviewer, we amended the figure legends to clarify the description for Student’s t-test and ANOVA used in each experiment.In Fig 1,  C refers to the replicon system and D to the infection as corrected by the reviewer.

Reviewer 2 Report

This is a very interesting study showing the role of the G protein coupled receptor, liraglutide, a Glp-2 agonist on intrahepatic signaling. The study is very well done. The authors have improved the quality of the figures in the revised manuscript.